# Detection and Modeling of Unstructured Roads in Forest Areas Based on Visual-2D Lidar Data Fusion

Guannan Lei [1,2], Ruting Yao [1,3], Yandong Zhao [1,2,4] and Yili Zheng [1,*]

1   School of Technology, Beijing Forestry University, Beijing 100083, China; guannanlei@bjfu.edu.cn (G.L.);
    yaoruting@bjfu.edu.cn (R.Y.); yandongzh@bjfu.edu.cn (Y.Z.)
2   Beijing Laboratory of Urban and Rural Ecological Environment, Beijing Municipal Education Commission,
    Beijing 100083, China
3   Key Laboratory of State Forestry Administration for Forestry Equipment and Automation,
    Beijing 100083, China
4   Research Center for Intelligent Forestry, Beijing 100083, China
*   Correspondence: zhengyili@bjfu.edu.cn

**Abstract:** The detection and recognition of unstructured roads in forest environments are critical for smart forestry technology. Forest roads lack effective reference objects and manual signs and have high degrees of nonlinearity and uncertainty, which pose severe challenges to forest engineering vehicles. This research aims to improve the automation and intelligence of forestry engineering and proposes an unstructured road detection and recognition method based on a combination of image processing and 2D lidar detection. This method uses the "improved SEEDS + Support Vector Machine (SVM)" strategy to quickly classify and recognize the road area in the image. Combined with the remapping of 2D lidar point cloud data on the image, the actual navigation requirements of forest unmanned navigation vehicles were fully considered, and road model construction based on the vehicle coordinate system was achieved. The algorithm was transplanted to a self-built intelligent navigation platform to verify its feasibility and effectiveness. The experimental results show that under low-speed conditions, the system can meet the real-time requirements of processing data at an average of 10 frames/s. For the centerline of the road model, the matching error between the image and lidar is no more than 0.119 m. The algorithm can provide effective support for the identification of unstructured roads in forest areas. This technology has important application value for forestry engineering vehicles in autonomous inspection and spraying, nursery stock harvesting, skidding, and transportation.

**Keywords:** unstructured road recognition; forest engineering; unmanned ground vehicle (UGV); autonomous navigation; multi-sensor fusion

## 1. Introduction

The role of unmanned operating vehicles in forest areas to promote sustainability and improve efficiency is fully recognized in the forest supply chain. The detection and recognition of unstructured roads in forest areas are of great significance for the autonomous navigation of forestry engineering vehicles. To solve the problem of unstructured road identification, many methods have been proposed.

The road model method is a common road-recognition method [1–4]. Xu Ming et al. used the two-dimensional Otsu algorithm to segment the region of interest and extract the road boundary [5]. Based on the judgment of the road model by the offline scene classifier, Zuo Wenhui et al. proposed that an off-line scene classifier was efficiently learned by both low- and high-level image cues to predict the unstructured road model [6]. The road model method has a significant effect on road recognition when there is a high level of structure. However, for completely unstructured roads with unclear edges, this method has several limitations [7–10].

Vanishing point detection is also a commonly used method for identifying unstructured roads [11–14]. Ende Wang proposed a method of light normalization based on RGB (Red, Green and Blue) images [15]. An online voting method based on the maximum weight was used to estimate the location of the vanishing point to determine the road boundary. Peyman Moghadam et al. used joint activities of four Gabor filters to estimate the local dominant orientation at each pixel location in the image plane [16]. The vanishing point was estimated by weighting the dominant direction of each pixel. However, in the forest area, a large number of trees and branches have a greater impact on the detection results of the straight-line classification [17,18].

With the development of hardware technology, the fusion of multiple sensors and multiple algorithms was used for UGV (unmanned ground vehicle) [19–21]. To overcome the low accuracy and poor robustness of a single detection method, a novel algorithm was proposed that combines an SVM (Support Vector Machine) and a K-nearest neighbors algorithm and the confidence map under a Bayesian framework [22]. This improved the overall road detection performance. A target detection scheme for an outdoor traversable area based on binocular vision was also proposed. Possible candidate pairs for road boundaries were selected from the region boundaries, and the Bayes rule was used to choose the most probable candidate pairs as the lane boundaries [23,24]. However, these methods use extremely complex algorithms, and their real-time performance needs to be improved.

In addition, the increasing interest in neural networks and deep learning stimulates new ideas for image segmentation and classification [25–28]. Luca Caltagirone et al. used a fully convolutional neural network for light detection and ranging lidar fusion to improve the reliability of unstructured environment recognition results [29]. A road-area detection method based on a Decision-Based Neural Network (DBNN) for outdoor robot single-camera navigation was also proposed [30]. However, the rapid calculation of deep learning and recognition of images relies highly on high-performance hardware support and a significant amount of supervised learning. This greatly increases the cost of equipment. The real-time performance needs to be further improved.

In summary, although much research has been conducted on road detection in complex outdoor environments, research on the detection of unstructured roads in forest areas is quite limited [31,32]. Most studies have only focused on the detection and annotation of road regions in images and have not provided accurate road models or spatial coordinate descriptions.

This study proposes an unstructured road detection system based on a vision camera and a two-dimensional (2D) lidar to address the problem of unstructured road detection and recognition in forest environments. Section 2 introduces a self-built forest autonomous navigation system and the environmental perception method. This section introduces an unstructured road-area recognition algorithm based on visual images. The algorithm adopts the strategy of "SEEDS + SVM" to meet the real-time requirements of the mobile platform. Then, combining the image information, the remapping method of the point cloud on the image and construction of the road model is introduced. Section 3 shows the experimental results of the algorithm in the forest area. Section 4 analyzes the effect of unstructured road recognition. Finally, a feasibility evaluation of the algorithm is conducted and discussed.

## 2. Materials and Methods

### 2.1. Autonomous Navigation Platform of Vision and Lidar Cooperation

For the UGVs to navigate in a forest environment, they need the ability to perform real-time road detection and direction estimation simultaneously [33,34]. Therefore, the UGV systems need to have the ability to comprehensively process the lidar point cloud data and visual images. This enables the vehicle and the system to understand the surrounding environment more accurately, as well as to realize the UGV's motion planning and obstacle avoidance.

In this study, a combination of a 2D lidar and a camera is selected as the forest road information perception system. During the experiment, a new external calibration method is used to realize online data space information matching between the vision camera and the 2D lidar. Figure 1a shows the 3D calibration board that we designed. Figure 1b shows the calibration test and calibration effect diagram of the calibration board in a laboratory environment. The matching transformation in the spatial dimension is realized by calculating the rotation–translation relationship between the vision and the laser system. Figure 1c shows the calibration of the vision sensor and lidar in the time dimension. The arrow in the figure indicates the time axis. Each circle or square on the timeline represents a data frame. The green graph connected by the dotted line indicates data with matching time-stamps. The remaining data (represented by the yellow graph) were discarded. The information flow is aligned along the time axis by extracting the message time-stamp. Detailed information regarding the calibration methods can be found in our previous study [35].

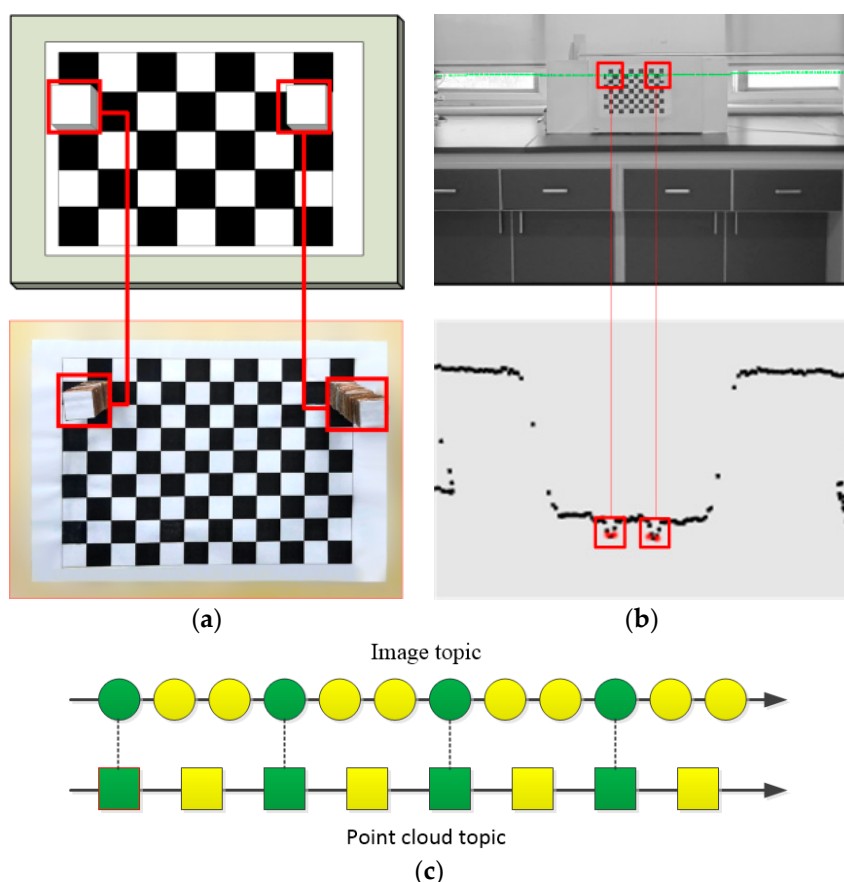

**Figure 1.** Joint calibration of the monocular vision camera and the 2D Lidar based on space coordinates and time axis conditions. (**a**) 3D calibration board; (**b**) The calibration test; (**c**) The calibration of the vision sensor and the lidar in the time dimension.

For meeting the needs of forest operations, as shown in Figure 2, the UGV platform used in the research is a self-built Ackerman chassis platform. It is equipped with CCD (charge coupled device) monocular vision camera (LRCP10230_1080P) and 2D Laser HOKUYO UST_10LX. The installation method between the sensors is vertical installation. The advantage of this configuration is that it neither obstructs the field of view but also facilitates the fusion of the two-sensor data.

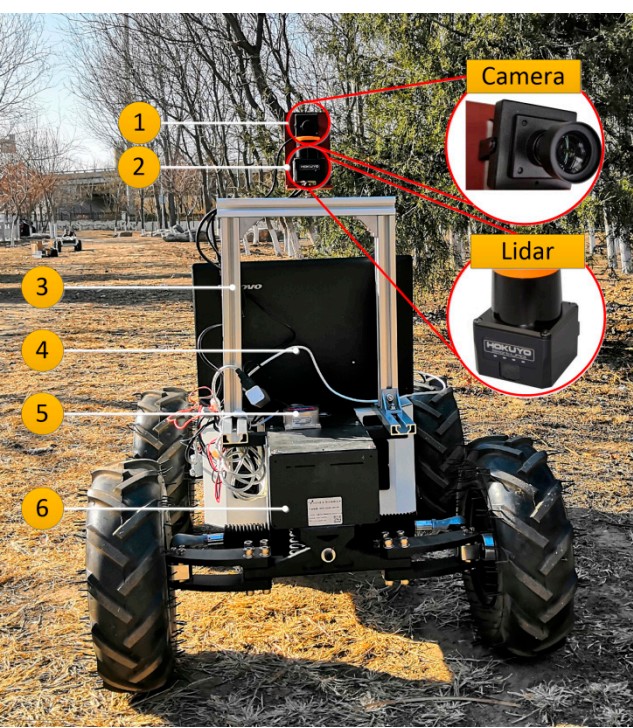

**Figure 2.** The self-built forestry UGV platform, equipped with a CCD monocular vision camera and a 2D lidar. 1. CCD monocular vision camera; 2. Two-dimensional lidar; 3. Sensor bracket; 4. Epigynous machine; 5. Gyro; 6. Ackerman chassis.

Based on the self-designed Ackerman chassis UGV test platform, the test environment that meets the conditions is configured. The epigynous machine is implemented on a PC with an Intel Duo Core i7-8700K CPU @ 3.7 GHz and 8 GB of RAM under Linux 16.04 and ROS-kinetic. Two-dimensional Lidar (HOKUYO UST_10LX) is selected to obtain point cloud data. The effective detection distance is 10 m. The scanning angle is 270°. The image collected by the CCD vision camera (LRCP10230_1080P) is 640 (pixels) × 480 (pixels).

The height of the lidar from the ground is 0.83 m. The height of the camera from the ground is 0.87 m. The experiments involved in this research are based on this UGV platform. The external parameters of the 2D lidar and the camera were calibrated, and the result was:

$$R = \begin{bmatrix} 0.0031445 & -0.99997 & -0.00714269 \\ -0.0234563 & 0.007067 & -0.9997 \\ 0.99972 & 0.00331109 & -0.0234334 \end{bmatrix} \tag{1}$$

$$T = \begin{bmatrix} -0.0146695 & 0.0222217 & -0.135979 \end{bmatrix}^T \tag{2}$$

### 2.2. Unstructured Road Detection Based on Vision

This section mainly introduces the image-processing methods and principles. To realize the rapid identification of roads in forest areas, the "SEEDS + SVM" strategy is adopted. Image processing is a key component of multi-sensor road recognition.

#### 2.2.1. Superpixel Segmentation

1. Preliminary segmentation;

It is challenging to extract and identify forest road areas owing to their non-structural features. To extract the road area from the image to guide the driving of the UGV, an improved SEEDS image segmentation algorithm is proposed.

The improved SEEDS algorithm first performs the initial segmentation of the image. The CCD monocular vision camera extracts the image of the forest area and reads it into

the system. The images are converted into the OpenCV Mat multi-channel image type. Then, depending on the pixel size of the image, the image is segmented. The image is initially divided into *K* rectangular sub-regions (superpixels) of the same size. The initial segmentation of the image is related only to the image size and space constraints. $N_{image}$ is the total number of pixels in the image, and *K* is the preset number of superpixels. The mapping relationship is as follows:

$$s(i) : \{1, \cdots, N_{image}\} \rightarrow \{1, \cdots, K\} \tag{3}$$

where $s(i)$ is the set of all pixels in the superpixel, and *i* is the pixel assigned to *s*.

As shown in Figure 3a, it is the original image of the road collected in the forest area. The number of initial segmentations is $K = 4 \times 4$ in Figure 3b. In Figure 3c, $K = 6 \times 5$. The larger the value of *K*, the higher the accuracy of the final segmentation result. However, the number of calculations has also increased. Therefore, in practical applications, to balance the calculation accuracy and time cost, the *K* value needs to be artificially set according to the size of the collected image and the configuration of computer hardware.

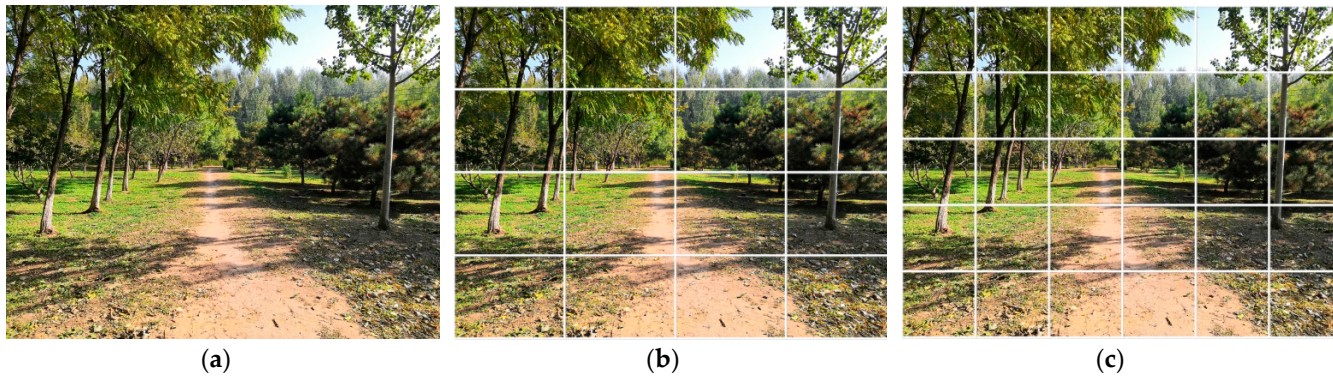

(**a**)　　　　　　　　　　　　　　　(**b**)　　　　　　　　　　　　　　　(**c**)

**Figure 3.** Superpixel initial segmentation effect diagram based on the image collected by the CCD monocular vision camera. (**a**) Raw image; (**b**) Initial segmentations with $4 \times 4$; (**c**) Initial segmentations with $6 \times 5$.

2. Energy function construction.

The superpixels of the initial segmentation are usually not the optimal segmentation results. The superpixel segmentation needs to be optimized based on the initial segmentation. The problem of image segmentation optimization is challenging. The reason is that the result of superpixel segmentation should be consistent in perception. In particular, the color should be as uniform as possible. However, this evaluation is subjective. In addition, there is no unified mathematical method for the best way of evaluating the uniformity of regional colors [36]. Almost every study on superpixels in the literature introduces a new energy function to satisfy its algorithm. To meet the unstructured road identification in the specific environment of the forest area, in this research, the energy function is constructed as follows:

$$E(s) = \alpha H(s) + \beta G(s) + \gamma CO(s) \tag{4}$$

The energy function is composed of three image-related indicators. The first term is $H(s)$ based on the color distribution concentration of the superpixels. The second term $G(s)$ is the boundary term. The third term is the compactness term $CO(s)$. The energy function is built by evaluating the color density distribution of each superpixel. $E(s)$ is the energy value of the evaluated superpixel.

3. Color distribution item $H(s)$;

The color distribution item is a superpixel evaluation index based on a histogram of the color distribution density. According to the color distribution in the superpixel, a histogram of the color distribution density is established:

$$H(\text{s}) = \sum_k \psi\left(c_{A_k}\right) \tag{5}$$

$$\psi\left(c_{A_k}\right) = \sum_{H_k} \left(c_{A_k}\right)^2 + \left(\text{g}\left(c_{A_k}\right)\right)^2 \tag{6}$$

$$c_{A_k}(j) = \frac{1}{S} \sum_{i \in A_k} \delta(I(i) \in H_k) \tag{7}$$

where $\psi\left(c_{A_k}\right)$ is a function that forces the color distribution to be concentrated in one or several colors; $c_{A_k}$ is the color histogram of the set of pixels in superpixel $A_k$; $\text{g}\left(c_{A_k}\right)$ is the difference coefficient; $j$ is the bin of the histogram; $I(i)$ represents the color of pixel $i$; $s$ is the normalization factor of the histogram; and, $\delta(\cdot)$ is the probability and statistical function equation. When the color of the pixel falls into the histogram, the value returned is 1; otherwise, the value returned is 0. $H_k$ is a closed subset of the color space.

$$\text{g}\left(c_{A_k}\right) = 1 - e\left(c_{A_k}\right) \tag{8}$$

where $e\left(c_{A_k}\right)$ is the information entropy. Information entropy is introduced to measure the degree of dispersion of the color distribution in the superpixels. The information entropy expression is described as:

$$e\left(c_{A_k}\right) = -\sum_{H_k} c_{A_k}(j) \ln\left(c_{A_k}(j)\right) \tag{9}$$

The color distribution in the superpixel color block is less concentrated as the information entropy increases. The color distribution item is formed by the color density histogram $\psi\left(c_{A_k}\right)$ and the color difference item $\text{g}\left(c_{A_k}\right)$. This item is used to measure the concentration of the superpixel color distribution and is the core term of the energy function.

4. Boundary term $G(s)$;

Based on the color distribution term, the boundary term is introduced to evaluate the smoothness of the superpixel boundary. It is used to remove local irregularities at the boundary of the superpixels. For superpixels with smoother borders, this term can improve the smoothness of the superpixel borders. However, the shape of the more popular borders seems to be subjective. The programmer can use this item to exercise fine control over the boundary of the superpixel.

With each boundary pixel as the center, a patch (of size $n \times n$) area $N_i$ is set around it. Similar to the color distribution item, the local color distribution item of each patch is counted depending on the color distribution of the histogram. The attribution of boundary pixels is determined by comparing the similarities between the main color distribution and the color distribution of the field. Similar to the color distribution item, the algorithm uses a quality measure to define $G(s)$:

$$G(s) = \sum_i \sum_k \left(b_{N_i}(k)\right)^2 \tag{10}$$

$$b_{N_i}(k) = \frac{1}{s} \sum_{i \in N_i} \delta(I(i) \in A_k) \tag{11}$$

where $k$ is the bin of the histogram. Each bin corresponds to a superpixel label. The histogram counts the number of pixels in the boundary patch that fall into label $k$. The boundary patch centered on the boundary pixel may belong to any superpixel adjacent to it.

It is generally believed that when most patches contain only one superpixel, the superpixel has a better shape.

5. Compactness term $CO(s)$.

Compactness (*CO*) is an index used to measure the compactness of the superpixels. The compactness term is specifically used to control the compactness of the superpixel and punish the irregular border of the superpixel.

$$CO(S_j) = \frac{4\pi A(S_j)}{P^2(S_j)}, \ (0 < CO \leq 1) \tag{12}$$

*CO* compares the area $A(S_j)$ of each superpixel $S_j$ with the area of a circle (the most compact 2D shape) with the same perimeter $P(S_j)$. The higher the value of *CO*, the better the compactness of the superpixel. The boundary term and the compactness term of the patch area $N_i$ are weighted by voting. When $G(s)$ has a large contribution to the energy function, the patch is considered to belong to superpixel $S_j$. When the weighted vote of the compactness term is greater than the maximum of the boundary term, the patch is eliminated from superpixel $S_j$. Thus, the purpose of boundary smoothing and superpixel compactness segmentation is achieved through the boundary penalty.

Based on constructing the energy maximization function, the Hill-Climbing algorithm is selected for optimization of the segmentation of superpixels. Superpixel optimization and updating is the process of maximizing the energy of a pixel block.

### 2.2.2. Road Detection Based on Online SVM

Based on the principle of energy maximization, the frame image is divided into multiple superpixel regions. Superpixels must be further semantically recognized and classified. The SVM is used for the online recognition of forest road areas.

1. Superpixel feature extraction;

Accurate and effective monitoring and classification of the divided superpixel area is a prerequisite for identifying forest roads.

To perform accurate semantic recognition of superpixels, the characteristics of different types of superpixels need to be analyzed and extracted. The image features include mainly color and texture features. Seven independent features (hue, saturation, value, angle second moment, contrast, inverse difference moment, and correlation) are selected to describe the superpixels to consider the accuracy of the classification and the time cost of feature extraction.

These characteristics are selected based on the specificity of the images collected in the special environment of the forest area. Considering the universality of tree canopy occlusion and shadows in the forest environment, the Hue, Saturation, and Value (HSV) color space performs better in adapting to shadows and reflections. Therefore, during the recognition process, the images are converted to HSV color space for classification. It has the advantage of removing interference for the area where there is a shadow projection of the canopy on the road in the forest area. In addition, there are many common elements such as trees, grasslands, and bushes in the forest environment. The strong texture features of these background environments provide important information for superpixel feature extraction. Considering the accuracy and time cost of classification, in the 14 texture feature statistics of the Gray Level Co-occurrence Matrix (GLCM) algorithm, four indicators are selected to quantitatively describe the texture features of superpixels. They are the angle second moment (*a*), Contrast (*c*), Inverse difference moment (*i*), and Correlation (*d*). For the training and supervised learning of the SVM model, a Boolean value is added to the feature operator as a superpixel classification label. The label for the non-road area is 0. The label of the road area is 1. Thus, the feature operator is represented as $\tau = [h, s, v, a, \ c, i, d, bool]^T$.

2. Construction and training of SVM model

- Construction of SVM model;

The superpixel feature operator, which is the basis for superpixel classification, is used as the input to the SVM model. The structures of the nonlinear positive and negative samples based on the superpixel feature operator are as follows:

$$\begin{cases} D = (\tau_i, y_i), i \in N_s \\ y_i \in \{1, -1\} \end{cases} \tag{13}$$

where $\tau_i$ is the input sample; $y_i$ is the category label of the sample; $N_s$ is the number of superpixels in a single frame; and, $D$ represents the training sample set. Because $\tau_i$ is a multidimensional sample space, the hyperplane of the classification model is

$$f(x) = w^T x + b \tag{14}$$

where $w$ is the normal vector, which determines the direction of the hyperplane; and $b$ is the displacement term, which determines the distance between the hyperplane and the origin. Then, a nonlinear mapping of $\tau_i$ is:

$$\tau_i \rightarrow \phi(\tau_i) = (\phi_1(\tau_i), \phi_2(\tau_i), \dots, \phi_n(\tau_i))^T \tag{15}$$

where $\phi(\tau_i)$ is the kernel function. After mapping, the optimal classification function model is:

$$f(\tau) = \sum_{i=1}^{n} \alpha_i y_i \phi(\tau_i)^T \cdot \phi(\tau) + b \tag{16}$$

Owing to the nonlinearity of the characteristic operator, it is necessary to ensure that the classification algorithm model has a good generalization ability and improves the calculation efficiency. The Gaussian radial kernel function (RBF) was selected for training in this study (as shown in Equation 17). The sample is mapped to a higher-dimensional space through the Gaussian kernel function, which makes the boundary of decision-making diverse and accurate, and has good calculation and classification performance.

$$\phi(\tau, \tau_i) = exp\left(-\frac{\| \tau - \tau_i \|^2}{\delta^2}\right) \tag{17}$$

Then, the training SVM model is tested using the test set. The trained SVM model can perform binary classification judgment on whether the superpixel is a road area. Thus, it can achieve superpixel classification and road-area recognition.

- Data set construction.

To enable the classification model to be fully trained and have a wide range of adaptability, a reasonable dataset needs to be constructed. In the process of supervised training of the SVM, a public dataset of Robot Unstructured Ground Driving (RUGD) is used [37]. The dataset contains more than 7000 frames of pixel-level images. The reason for selecting this dataset is that it is different from the existing benchmark data for autonomous driving. It contains more terrain types, irregular boundaries, minimally structured scales, fewer urban elements, and structured road elements. In addition, there are many unstructured roads in the forest environment. Among these, 850 images containing forest roads are selected. Then, 1100 images of the forest environment taken by our laboratory were also added to the dataset. The shooting location is the Jiufeng National Forest Park and the Liaocheng Forest Farm Test Base, Shandong, China. They formed a dataset containing 1950 frames of images.

A total of 1300 images were randomly selected as the training set for the SVM. Special attention needs to be paid to the differences between the training set and the training sample. The samples input to the SVM for training were superpixel feature operators. In this study, each image was segmented into 5 × 5 superpixels using the improved SEEDS

algorithm. Therefore, the number of training samples used for SVM training was 25 times the number of images, that is, 32,500. It contained a positive sample of 9053 and a negative sample of 23,447. The superpixels in the road area are positive samples that include road surfaces with dirt, gravel-covered road surfaces, humus-covered road surfaces and road surfaces with vegetation projections. Superpixels in non-road areas are negative samples that include the sky, tree trunks, branches, weeds, and bushes. The establishment of the dataset provides an important foundation for the training and testing of the SVM model. The detailed results of the online test of the SVM model are presented in Section 2.3.

### 2.3. Description of Unstructured Road Structure Based on 2D Lidar

The 2D lidar can obtain environmental information and compensate for the lack of a visual camera [38]. Therefore, a 2D lidar was mounted on the test platform. The 2D point cloud can be used to detect the azimuth and width information of the passable area between the forests. In addition, it can be used to measure the distance relationship between the UGV and the obstacles.

#### 2.3.1. Lidar Point Cloud Acquisition and Processing

The acquisition and processing of point cloud data is also a key link in unstructured road recognition. Owing to the complex environment of the forest area, the amount of point cloud data obtained is relatively large. Therefore, the point cloud data must be filtered. The purpose of filtering is to filter out a large amount of interference information in the point cloud data, and it also facilitates the matching of the radar point cloud and the visual image.

Straight-through filtering is an effective point cloud processing method. It trims the point cloud within a specific coordinate range by specifying the interval range of each coordinate axis to obtain the target point cloud. The processing method for the 2D point cloud through filtering is:

$$\begin{cases} x_{min} - \Delta_x \leq x_i \leq x_{max} + \Delta_x \\ y_{min} - \Delta_y \leq y_i \leq y_{max} + \Delta_y \end{cases} \tag{18}$$

where $(x_i, y_i)$ are the point cloud coordinates. In the forest environment, the main information reflected by the 2D point cloud is the position and shape of the tree-level bushes on both sides of the unstructured road. Point clouds that are too far away have limited significance for the autonomous navigation of UGVs. In this research, $x_{min} = -3$ m, $x_{max} = 3$ m, $y_{min} = -3$ m, $y_{max} = 9$ m, $\Delta_x = \Delta_y = 0.5$ m.

As shown in Figure 4, the red points represent the point clouds detected by lidar. The blue dot represents the current position of the UGV. The sector is the lidar viewing angle. The light green area is the range of thorough filtering. The point cloud in the non-light green area is discarded, and only the point cloud after pass-through filtering is retained. Therefore, after the point cloud data are processed, the area indicated by the white arrow is the passable area. Only the point cloud on both sides of the current road is left. There were a total of 1080 original point cloud data. After filtering, approximately 85 data remain per frame.

#### 2.3.2. Remap Transformation

The 2D point cloud boundary generated by the 2D lidar can obtain the depth and width information of the environment, and the lidar can reasonably describe the parameters of the target. However, the distance between the trees on both sides of the road does not accurately define the road boundary.

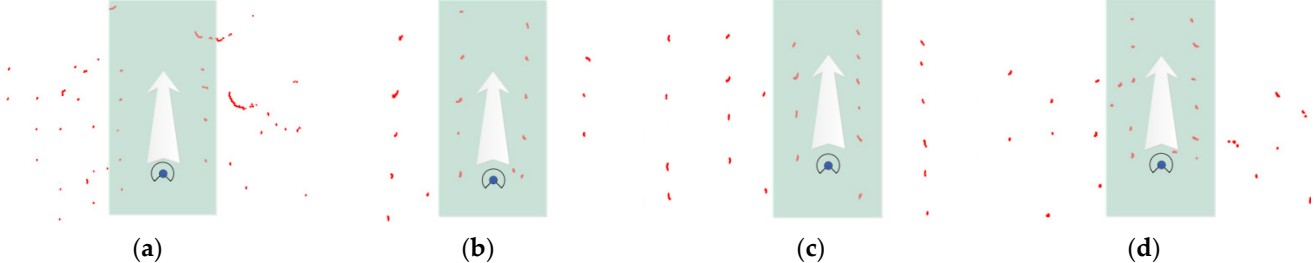

**Figure 4.** 2D lidar point cloud autonomous navigation system orientation in different forest scene. (**a**) Point cloud 1; (**b**) Point cloud 2; (**c**) Point cloud 3; (**d**) Point cloud 4.

The calibrations of the camera and 2D lidar mentioned above (in Section 2.2) are the external calibration parameters obtained under the joint calibration experiment after the equipment was installed. On this basis, the image information and point cloud information are remapped. The filtered 2D point cloud is remapped onto the image, and then the coordinate ratio of the two boundaries on the image is used to find the approximate width of the road. The principle of the algorithm is illustrated in Figure 5.

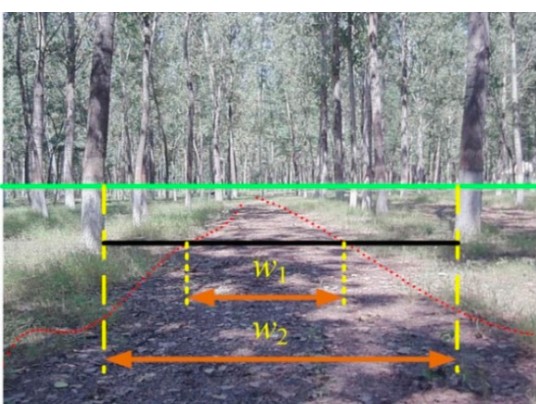

**Figure 5.** Schematic of remapping relationship.

The red dashed line represents the road boundary fitted to the image. The solid green line is the horizontal line scanned by lidar. $w_2$ is the distance between two obstacles in front of the vehicle based on the 2D laser point cloud measurement. $w_1$ is the width of the road boundary in the image. $p_1$ is the number of pixels corresponding to $w_1$. $p_2$ is the number of pixels corresponding to $w_2$.

According to the proportional equation, $w_2$ can be obtained:

$$\frac{w_1}{w_2} = \frac{p_1}{p_2} \tag{19}$$

## 3. Experimental Results and Analysis

This section introduces the experimental results. The feasibility and effectiveness of the proposed method are analyzed and explained through the establishment of the road model.

### 3.1. Experimental Environment Configuration

The experiment was conducted in a wild forest environment. The image data are from the dataset mentioned above (Section 2.2.2). The experiment considered the differences in the appearance of vegetation caused by seasonal changes. The dataset also contains samples from different seasons of winter and summer to ensure the diversity and completeness of the data samples. Notably, there are no corresponding laser point cloud data in the RUGD public data set. The road fitting related to the laser point cloud data is all the

experimental data obtained from Jiufeng Forest Farm of Beijing Forestry University and Liaocheng Forest Farm Test Base in Shandong, China. However, the images obtained from the RUGD dataset can still be used to test and verify the effectiveness of the proposed image-processing method.

### 3.2. Visual Image Processing

3.2.1. Superpixel Segmentation in Real-Time

In the process of superpixel segmentation, the initial number of segmentations *K* is set by the programmer according to the actual situation. In practical applications, to balance the calculation accuracy and time cost, it is necessary to set the *K* value according to the size of the collected image and the configuration of the computer hardware. In this experiment, $K = 5 \times 5$.

To ensure the validity and reliability of the algorithm, we collected as many different roads as possible. Figure 6 shows the effect of the superpixel segmentation. The number (0–24) in each image in the figure is the label of each superpixel. Figure 6a–f are from the RUGD public dataset. Figure 6g–k are from the Jiufeng Forest Farm of Beijing Forestry University. Figure 6l–p are from the Shandong Liaocheng Forest Farm Test Base. Figure 6a–k are summer road scenes, and Figure 6l–p are spring and autumn road scenes. The figures contain a variety of unstructured roads in forest areas, where Figure 6a,c,f are wet dirt roads. Figure 6l–n are dry dirt roads. Figure 6b,d are gravel roads. There are shadows of trees on the road in Figure 6i,j. There are mostly weeds on the roads in forest areas in Figure 6a,e,g,h,i,k. There is a lot of humus on the road and roadside in Figure 6m,p. Figure 6 shows that the improved SEEDS algorithm proposed in this study can segment the road and non-road areas accurately and has a good superpixel segmentation effect under various complex road conditions.

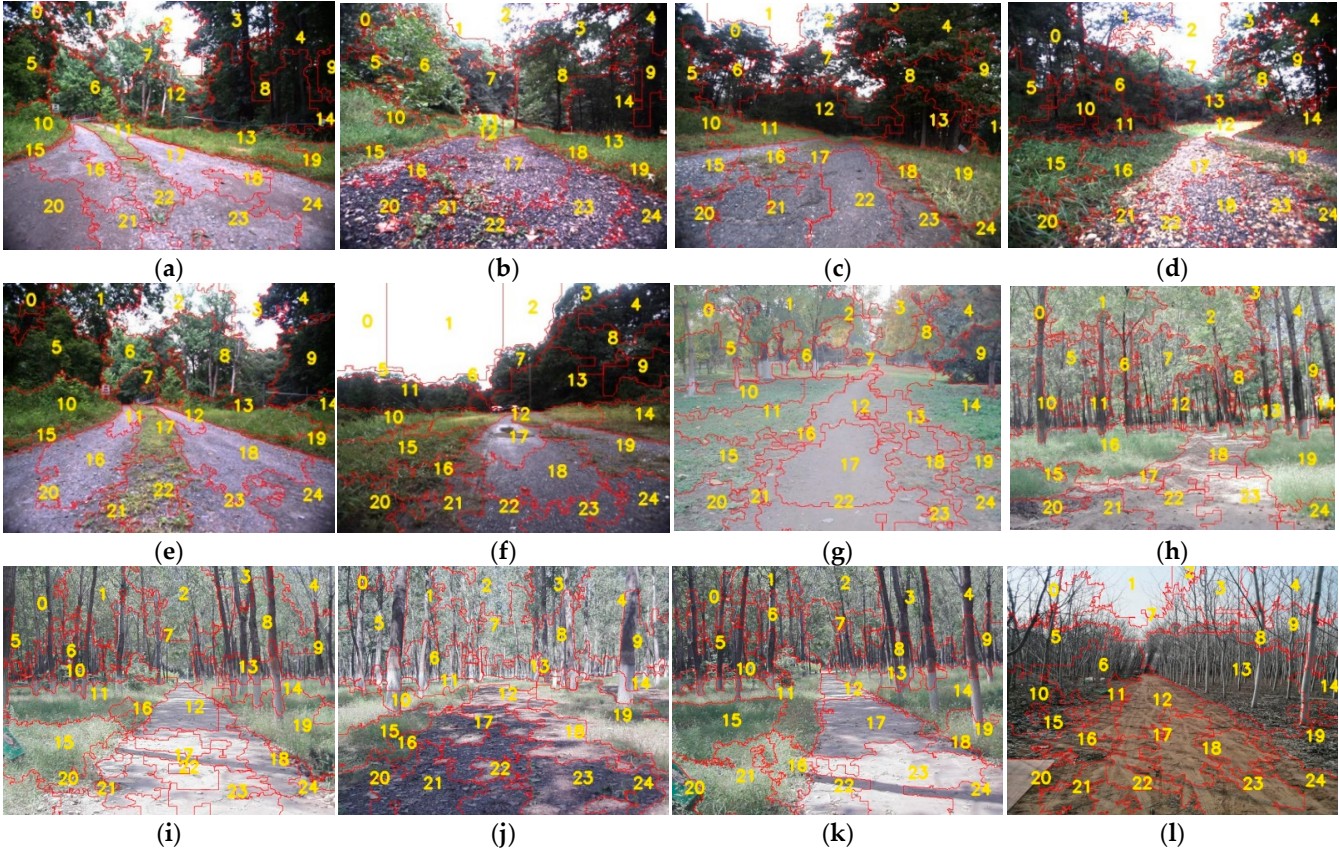

**Figure 6.** *Cont.*

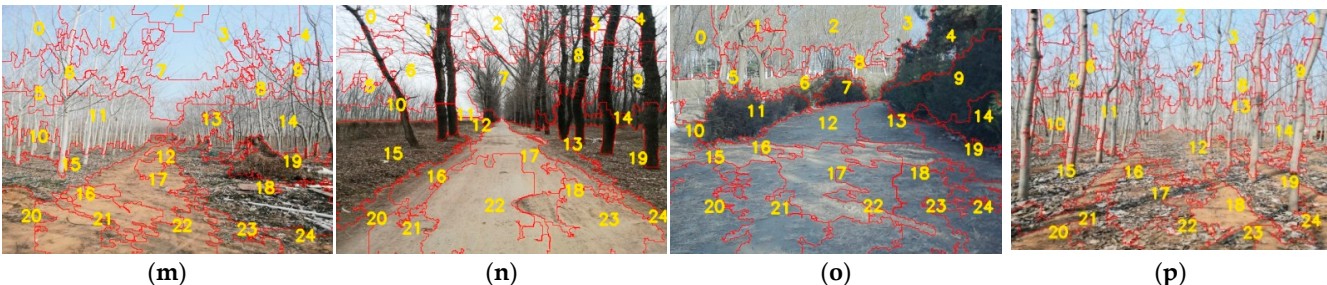

**Figure 6.** Superpixels segmentation and calibration results of unstructured roads based on improved SEEDS algorithm in forest areas. (**a–f**) are from RUGD; (**g–k**) are from Jiufeng Forest Farm; (**l–p**) are from Liaocheng Forest Farm.

### 3.2.2. Online Recognition of Road Area

Based on superpixel segmentation, the feature operator of superpixel $\tau$ is extracted. The trained SVM model classifies whether the superpixel represents the road area according to the feature operator of the superpixel in real-time. Binarization is performed on classified superpixels. As shown in Figure 7, the road area is marked in blue, and the non-road area is marked in black.

**Figure 7.** SVM-based binary images of road areas and non-road areas, and extraction of irregular road boundaries. (**a–f**) are from RUGD; (**g–k**) are from Jiufeng Forest Farm; (**l–p**) are from Liaocheng Forest Farm.

The images in Figure 7 correspond to the images in Figure 6. As shown in Figure 7, the shape and boundary of the superpixels are irregular. Owing to the influence of light or grass plants on the road, the non-road area and road area may extend mutually during the superpixel segmentation process. As a result, some incorrectly classified patches may appear in the SVM detection results. These patches have an impact on area division and road boundary fitting. Therefore, open operation in morphology is used to eliminate the patch caused by the classification error in the road area. In this way, the road and non-road areas will be separated.

The road boundary was obtained using the Canny edge detection algorithm. Then, by obtaining the real outline of the road image, the road boundary is extracted. We use the position of the pixel in the image coordinates to find the farthest point on the contour as the dividing point. The left and right boundary separation results are presented in Figure 7. The red and green polylines are the left and right borders, respectively.

Unstructured road areas and boundaries were also obtained. The detection and recognition of unstructured road areas in forest areas is a prerequisite for establishing road models.

### 3.2.3. Road Model Establishment

The boundary of the unstructured roads in the forest area is very irregular. To use road recognition results for UGV autonomous navigation, an irregular road boundary needs to be fitted and a road model established. The extraction of the road boundary provides a basis for the establishment of the road model. A polynomial is used to fit the point set of the left and right boundaries. The error function is:

$$L\varepsilon = \sum_{i=0}^{nl} \left[ L(x_i) - LX^i \right]^2 \tag{20}$$

$$R\varepsilon = \sum_{i=0}^{nr} \left[ R(x_i) - RX^i \right]^2 \tag{21}$$

When the error function is not greater than the preset minimum value, the fitting requirements are met, and the fitting curve equations of the left and right boundaries are obtained. The results of the final fitted smooth road boundary curve are shown in Figure 8. Figures 7 and 8 show one-to-one correspondence.

The red curve in Figure 8 is the road boundary line. The road between the two boundary lines is considered a passable area. The black curve represents the middle line of the road. After obtaining the road model, the matching experiment of the image and point cloud data is shown in Figure 8g–p. The sampling point coordinates in the UGV coordinate system are obtained.

The road model not only includes the road boundary line and the road centerline but more importantly, the conversion of an unstructured road to a structured road model. This is of great significance for the navigation of the UGV. The road boundary line explicitly defines road and non-road areas. The centerline of the road can guide the UGV away from the road boundary, which improves driving stability and safety. Moreover, in the forest environment, the smoothness of the road boundary is poor, and the centerline of the road is obtained by averaging the sampling points of the boundary. This method plays a role in mean filtering for the construction of the road centerline. Therefore, the centerline of the road is smoother than the boundary line. The road centerline can be directly used for the autonomous navigation of a UGV.

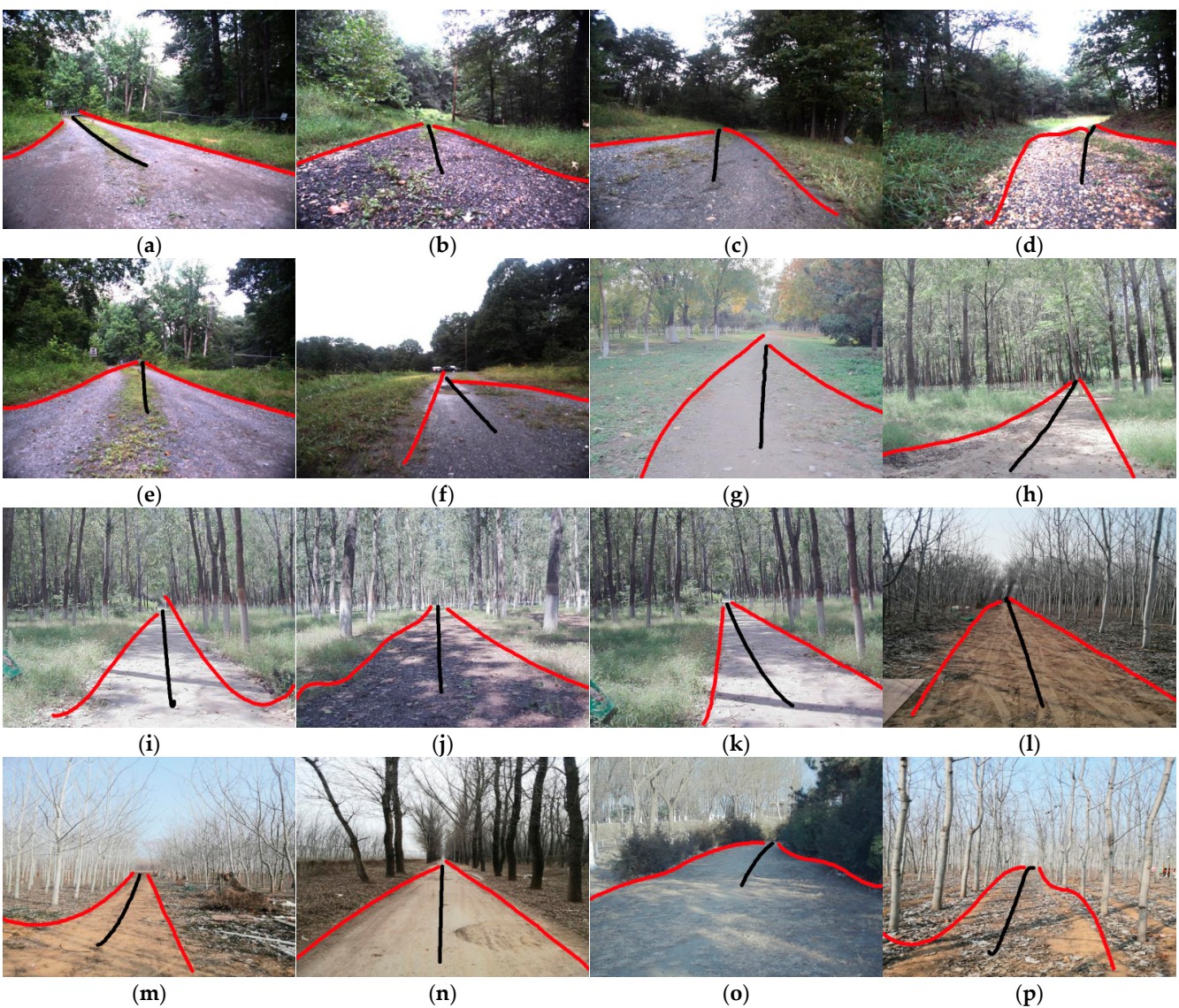

**Figure 8.** Smooth boundary fitting and road centerline fitting of unstructured roads in forest areas. (**a**–**f**) are from RUGD; (**g**–**k**) are from Jiufeng Forest Farm; (**l**–**p**) are from Liaocheng Forest Farm.

## 4. Discussion

### 4.1. Road Structure Evaluation

Based on the road model, the effect of road boundary fitting is further evaluated. Here, we first evaluate the road model from the perspective of image segmentation. Generally, based on the results of manual calibration, the results of the algorithm are compared with the results of manual calibration to evaluate the image segmentation effect of the algorithm. A total of 300 images are randomly selected from the test set of the three data sources mentioned above. These images were manually calibrated by the annotation tool "LableMe". The calibration results are compared with the algorithm fitting results. The comparison is shown in Table 1.

**Table 1.** Evaluation of unstructured road models in forest areas from the perspective of image segmentation.

| Data Sources | MIoU | | Precision | | Recall | | F1 | | MAE | |
|---|---|---|---|---|---|---|---|---|---|---|
| | SLIC | Improved SEEDS | SLIC | Improved SEEDS | SLIC | Improved SEEDS | SLIC | Improved SEEDS | SLIC | Improved SEEDS |
| RUGD | 0.8649 | 0.9005 | 0.8704 | 0.9097 | 0.9346 | 0.9618 | 0.9013 | 0.9332 | 0.0598 | 0.0301 |
| Jiufeng National Forest Park | 0.8762 | 0.9028 | 0.8843 | 0.9125 | 0.9398 | 0.9746 | 0.9112 | 0.9425 | 0.0465 | 0.0297 |
| Liaocheng Forest Farm | 0.8835 | 0.9173 | 0.8961 | 0.9241 | 0.9502 | 0.9795 | 0.9224 | 0.9526 | 0.0328 | 0.0216 |

In Table 1, the method proposed in this study is compared with the Simple Linear Iterative Cluster (SLIC) superpixel segmentation method. Here are five indicators selected to evaluate the effect of road segmentation from multiple aspects. It can be seen from the table that the improved SEEDS algorithm is better than the SLIC algorithm in all indicators. Based on the improved SEEDS, MIoU is above 90% and an F1 is above 93%. More importantly, during the experiment, it found that the average time for the SLIC algorithm to process each frame of image is about 700 ms. This is unbearable for systems with high real-time requirements. The algorithm proposed in this article means that the average processing time of each frame of image is 89.3 s. In addition, as can be seen, the statistical results of Shandong Forest Farm are slightly better than the other two. The reason may be that the logging road of Shandong Forest Farm is undergoing logging operations during the test period, and the road boundary is clearer and more recognizable. Therefore, it is more accurate in manual labeling and algorithm recognition. Considering the unstructured characteristics of forest roads, Table 1 shows that the road model established by the algorithm matches well with the road area manually calibrated.

The matching between the corresponding points of the image and the point cloud provides an important guarantee for the parameterization of the road model. Therefore, it is necessary to further analyze the road detection error under the remapping relationship according to the experimental results. The analysis of the remapping errors is shown in Table 2. Figure 8g–p are used for statistical analysis. Taking the abscissa of the sampled image as the benchmark, a sampling point is taken every 20 pixels on the boundary fitting curve. Through remapping, the position coordinates in the UGV coordinate system corresponding to the sampling points are determined.

**Table 2.** Remapping error analysis.

| Distance | $Eu_p$ (pixel) | $Ev_p$ (pixel) | $Eu_d$ (m) | $Ev_d$ (m) | $E_\xi$ (m) |
|---|---|---|---|---|---|
| 1 m–2 m | 25.4572 | 3.1517 | 0.0833 | 0.0086 | 0.0390 |
| 2 m–3 m | 28.3554 | 3.6740 | 0.1309 | 0.0127 | 0.0584 |
| 3 m–4 m | 32.3748 | 4.0512 | 0.1971 | 0.0221 | 0.0831 |
| 4 m–5 m | 35.9710 | 4.3476 | 0.2564 | 0.0272 | 0.1190 |

$Eu_p$ refers to the U-axis pixel average error between the road boundary pixel sampling point and the corresponding point cloud. $Ev_p$ refers to the V-axis pixel average error between the road boundary pixel sampling point and the corresponding point cloud. Table 2 shows that the horizontal average error and the vertical average error of the boundary sampling points both increases with increasing distance. However, in comparison, the U-axis error of the road boundary is greater. In the control process, the influence of the lateral error on the heading angle is particularly obvious. It also has a greater impact on the lateral stability and safety of vehicles. To reflect the fitting error more intuitively, $Eu_d$ and $Eu_p$ are introduced. $Eu_d$ and $Eu_p$ refer to the actual errors corresponding to $Eu_p$ and $Ev_p$, respectively.

In the process of UGV tracking control, the management of the previous steps has the most significant impact on the adjustment of the UGV attitude. During the control process, the road model and control parameters change dynamically. In addition, the roads in forest areas are uneven, and UGVs are generally traveling at low speeds ($v < 1.5$ m/s). A long-distance prediction has a limited impact on system movement. This causes unnecessary memory occupation and increases the computing costs. Therefore, the error analysis range is kept within 5 m of the front of the UGV. During the experiment, we measured the error in each interval at intervals of 1 m. It can be seen from Table 2 that the remapping error increases with the increasing distance.

In Table 2, the centerline of the road is the actual trajectory of the UGV tracks. $E_\xi$ is introduced, which refers to the lateral average error of the sampling points of the road centerline. The road centerline is obtained by averaging the sampling points corresponding to the left and right boundaries. This is equivalent to implementing mean filtering and effectively reduces the fitting error of the road centerline. The maximum average error of the road centerline within 5 m is 0.119 m. The above-mentioned fitting road width is in the range of 2–3 m. The average relative error does not exceed 6%. The image and radar data, therefore, match well. The road model can meet positioning and tracking requirements.

*4.2. Algorithm Real-Time Evaluation*

In addition to measurement accuracy, algorithm real-time performance is another important indicator of system performance.

For the improvement of SEEDS algorithm, this study reconstructs the energy function in the algorithm to adapt to image segmentation in a specific environment of the forest area. The boundary terms and the compactness term are introduced to properly control the compactness of the superpixels. At the same time, the superpixel segmentation effect and convergence speed are improved.

The boundary and compactness terms are essentially controlled by the superpixel boundary penalties, but they are implemented in different ways. The boundary term adjusts the boundary by selecting a superpixel boundary patch. The compactness term controls the overall shape of a superpixel using geometric constraints. In principle, this method has a wide range of adaptability. In this study, in a specific forest environment, the environmental features such as branches, trunks, and weeds in the collected images have irregular boundaries. To ensure the true road boundary as much as possible, these two weights were set to be small. The construction of the energy function provides an important guarantee for the fast and accurate segmentation of superpixels.

In order to further ensure the real-time performance of the system, the superpixels are not directly used as input parameters. Instead, superpixels are described by feature operators composed of features with eight dimensions. It is used as the input to train and test the SVM. Through the training of a large number of positive and negative samples, the SVM model can quickly realize the classification of superpixels and the discrimination of road areas in the actual working environment. On the other hand, given the characteristics of fewer people, cars, and houses in the forest areas, this research only focuses on the acquisition of road areas and does not specifically distinguish the specific semantics and components of non-road areas. Under the system configuration described above, the average time for the system to process each frame of the image is 89.3 ms. This can meet the basic requirements of the autonomous navigation platform to process forest scene images at 10 frames/s.

## 5. Conclusions

This study aims to improve the automation and intelligence of forestry engineering and proposes an unstructured road detection and recognition method based on a combination of image processing and lidar. The challenge of road recognition in forest areas is that, on the one hand, it lies in the road area recognition and real-time requirements in the complex environment of forest areas; on the other hand, it is the construction of reasonable

road models that can be truly used for unmanned vehicle navigation. This research has made great efforts to this end.

Forest roads are unstructured without obvious road boundaries. Because forest roads lack effective reference objects and manual signs, they have high degrees of nonlinearity and uncertainty. Aiming at the complex and rapidly changing forest environment, the driverless navigation system needs to quickly and accurately identify the surrounding road environment. Therefore, the unmanned vehicle system's understanding of the collected images is the primary problem faced by this research. During the experiment, it was found that image processing consumes considerable computing power. Therefore, this study adopts the strategy of "SEEDS + SVM" to achieve rapid image processing. This strategy basically meets the real-time requirements and accuracy requirements of the system for environmental recognition.

When a new sample appears, the upgrade and update of the training system can be completed by adding the new sample to the training set to train and update the model. This means that the system has good plasticity, robustness, and portability.

The images collected by the camera can reflect the color and texture characteristics of the surrounding environment but cannot accurately reflect the location information about objects in the environment. For an unmanned driving platform that navigates under complex outdoor scenes in forest areas, it is necessary to have the ability to perform real-time road detection and direction estimation simultaneously. Therefore, this study uses a combination of a visual camera and a 2D lidar to perceive and detect forest roads. Through remapping, the point cloud data and the corresponding points of the image data are matched. The road boundary and road centerline that incorporates depth information is directly converted into the vehicle coordinate system through coordinate conversion. The road centerline containing coordinate information can provide a good tracking path basis for autonomous mobile platforms. The centerline of the road can guide vehicles as far as possible from the border to ensure the safety and reliability of trajectory tracking. Therefore, the advantage of this research is also reflected in the establishment of a parametric road model based on the vehicle coordinate system, which greatly improves the autonomy and safety of UGV.

In response to the needs of autonomous and intelligent forest operations, this research realizes the detection and identification of unstructured roads in forest areas. The algorithm has good real-time performance, strong robustness and environmental adaptability. It can provide a more accurate parametric road model for UGV navigation in forest areas. It provides an important basis for environmental perception and autonomous navigation of a UGV in a forest environment. In the field of forestry engineering, this technology has important applications in autonomous inspection, skidding, transportation, autonomous spraying, and nursery stock tending.

**Author Contributions:** Conceptualization, G.L.; methodology, G.L.; software, G.L.; validation, G.L. and R.Y.; formal analysis, R.Y.; investigation, R.Y.; data curation, G.L.; writing—original draft preparation, G.L.; writing—review and editing, Y.Z. (Yili Zheng); visualization, R.Y.; supervision, Y.Z. (Yandong Zhao); project administration, Y.Z. (Yandong Zhao); funding acquisition, Y.Z. (Yili Zheng). All authors have read and agreed to the published version of the manuscript.

**Funding:** This research was funded by "the Fundamental Research Funds for the Central Universities", grant number 2021ZY74" and "the National Natural Science Foundation of China", grant number 31670719.

**Conflicts of Interest:** The authors declare no conflict of interest.

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
