# Peer review of "Detection and Modeling of Unstructured Roads in Forest Areas Based on Visual-2D Lidar Data Fusion"

_forests, doi:10.3390/f12070820_

Round 1
Reviewer 1 Report
The paper submitted for evaluation presents the results of research on improving automation and intelligence of forest engineering and proposes a non-structural method for road detection and recognition based on a combination of image processing and lidar. The challenge of road recognition in forest areas is firstly to detect road surfaces and real-time requirements in a complex forest environment, and secondly to create meaningful road models that can actually be used for navigation in unmanned vehicles. Although it is not only about navigation. From the reviewer's point of view, it would also be important to be able to perform automated inventory of forest road networks with such solutions.
Full-text evaluation requires advanced knowledge in mathematics and physics (especially research methods and analysis). Nevertheless, one gets the impression that the elaborated description is meticulous and exhaustive, which confirms the high value of the publication given the importance of the problem. In the reviewer's opinion, it makes a significant contribution to the development of methods for the detection and recognition of unstructured roads. I wish the authors many more ideas to increase the precision and speed of operation of the devices in this field.
Notes:
Humus roads - this is not a type of road pavements, the term here refers to all forest roads that are permanently or periodically covered with organic material, regardless of the type of road surface (concrete roads, asphalt roads, gravel roads, dirt roads ect.). Similarly "Shaded roads".
It is advisable to specify the number of road sections studied: "... To ensure the validity and reliability of the algorithm, we collected as many different roads as possible ...", i.e. how many, 16? - how can we conclude from the content of figure 6?
Author Response
Dear Reviewer,
We gratefully thank you for your time spend making your constructive remarks and suggestions, which has significantly arise the quality of the manuscript and has enable us to improve the manuscript. Each suggested revision and comment, brought forward was accurately incorporated and considered.
Original Manuscript ID: forests-1252668
Original Article Title: “Detection and modeling of unstructured roads in forest areas based on Visual-2D Lidar data fusion”
We are uploading (a) our point-by-point response to the comments (below) (Response to Reviewers), (b) an updated manuscript with “Track Changes” function for highlighting (Manuscript with Track Changes), and (c) a clean updated manuscript without highlights (Manuscript). Please see the attachments. We hope this revised manuscript has addressed your concerns, and look forward to hearing from you.
Thank you again for your valuable comment.
Best regards,
Prof. Yili Zheng
Dr. Guannan Lei
Data: June 11, 2021
School of Technology, Beijing Forestry University
Beijing, 100083, P.R.China
E-mail: guannanlei@bjfu.edu.cn
E-mail: zhengyili@bjfu.edu.cn

Reviewer 2 Report
General comments
This article presents a proposal for the detection and recognition of unstructured roads in the forest environment. The topic of the paper is highly topical, in forestry the above forms of work are beginning to be promoted. This research aims to improve the automation and intelligence of forest engineering. The article presents a method of path detection and recognition based on a combination of image processing and 2D lidar detection, which is also confirmed by experimental results and discussions. Here are some comments.
The introduction
The introduction offers a comprehensive overview of general knowledge in the field of fuel consumption in deforestation. It provides an adequate overview of the presented issues.
Materials and methods
Process chapters 2 to 4 as chapter material and methods. This part of the article is incoherent.
Experimental Results and Analysis
In the results section, in addition to the described results of published research, it would be appropriate to provide a discussion and comparison with already existing similar works.
Conclusions
More hypotheses are given. The presented results presented in the conclusion can be moved to the dissertation chapter. Such an extensive conclusion is not necessary.
Review Summary:
It needs to be modified according to the comments above.
Author Response
Dear Reviewer,
We gratefully thank you for your time spend making your constructive remarks and suggestions, which has significantly arise the quality of the manuscript and has enable us to improve the manuscript. Each suggested revision and comment, brought forward was accurately incorporated and considered.
Original Manuscript ID: forests-1252668
Original Article Title: “Detection and modeling of unstructured roads in forest areas based on Visual-2D Lidar data fusion”.
We are uploading our point-by-point response to the comments (below) . Please see the attachment. We hope this revised manuscript has addressed your concerns, and look forward to hearing from you.
Thank you again for your valuable comment.
Best regards,
Prof. Yili Zheng
Dr. Guannan Lei
Data: June 11, 2021
School of Technology, Beijing Forestry University
Beijing, 100083, P.R.China
E-mail: guannanlei@bjfu.edu.cn
E-mail: zhengyili@bjfu.edu.cn
